# Flowification: Everything is a Normalizing Flow

**Bálint Máté**[*]
University of Geneva
balint.mate@unige.ch

**Samuel Klein**[*]
University of Geneva
samuel.klein@unige.ch

**Tobias Golling**
University of Geneva
tobias.golling@unige.ch

**François Fleuret**
University of Geneva
francois.fleuret@unige.ch

## Abstract

The two key characteristics of a normalizing flow is that it is invertible (in particular, dimension preserving) and that it monitors the amount by which it changes the likelihood of data points as samples are propagated along the network. Recently, multiple generalizations of normalizing flows have been introduced that relax these two conditions [1, 2]. On the other hand, neural networks only perform a forward pass on the input, there is neither a notion of an inverse of a neural network nor is there one of its likelihood contribution. In this paper we argue that certain neural network architectures can be enriched with a stochastic inverse pass and that their likelihood contribution can be monitored in a way that they fall under the generalized notion of a normalizing flow mentioned above. We term this enrichment *flowification*. We prove that neural networks only containing linear and convolutional layers and invertible activations such as LeakyReLU can be flowified and evaluate them in the generative setting on image datasets.

## 1 Introduction

Density estimation techniques have proven effective on a wide variety of downstream tasks such as sample generation and anomaly detection [3–8]. Normalizing flows and autoregressive models perform very well at density estimation but do not easily scale to large dimensions [8–10] and have to satisfy strict design constraints to ensure efficient computation of their Jacobians and inverses. Advances in other areas of machine learning cannot be utilized as flow architectures because they are not typically seen as being invertible; this restricts the application of highly optimized architectures from many domains to density estimation, and the use of the likelihood for diagnosing these architectures.

Methods using standard convolutions and residual layers for density estimation have been developed for architectures with specific properties [11–14]. These methods do not provide a recipe for converting general architectures into flows. There is no known correspondence between normalizing flows and the operations defined by linear and convolutional layers.

In this paper we show that a large proportion of machine learning models can be trained as normalizing flows. The forward pass of these models remains unchanged apart from the possible addition of uncorrelated noise. To demonstrate our formulation works we apply it to fully connected layers, convolutions and residual connections.

The contributions of this paper include:

---

[*]Equal contribution.

- In §3.1 we show that linear layers induce densities as augmented normalizing flows [2] with the multi-scale architecture used in RealNVPs [6]. We also show how these layers can be viewed as funnels [15] to increase their expressivity. We term this process *flowification*.

- In §3.2 we argue that most ML architectures can be decomposed into simple building blocks that are easy to flowify. As an example, we derive the specifics for two dimensional convolutional layers and residual blocks.

- In §4 we flowify multi-layer perceptrons and convolutional networks and train them as normalizing flows using the likelihood. This demonstrates that models built from standard layers can be used for density estimation directly.

## 2   Background

**Normalizing flows**

Given a base probability density $\{p_0(z)|z \in Z\}$ and a diffeomorphism $f : X \to Z$, the pullback along $f$ induces a probability density $\{p(x)|x \in X\}$ on $X$, where the likelihood of any $x \in X$ is given by $p(x) = p_0(f(x))|\det(J_x^f)|$, where $J_x^f$ is the Jacobian of $f$ evaluated at $x$. Thus, the log-likelihoods of the two densities are related by an additive term, which will be referred to as the *likelihood contribution* $\mathcal{V}(x, z)$ [1]. Normalizing flows [3] parametrize a family $f_\theta$ of invertible functions from $X$ to $Z$. The parameters $\theta$ are then optimized to maximize the likelihood of the training data. A lot of development has gone into constructing flexible invertible functions with easy to calculate Jacobians where both the forward and inverse passes are fast to compute [6-8, 16, 17].

As the function $f$ must be invertible it is required to preserve the dimension of the data. This limits the expressivity of $f$ and makes it expensive to model high dimensional data distributions. To reduce these issues several works have studied dimension altering variants of flows [2, 1, 18-21].

**Dimension altering generalizations of normalizing flows**

**Reducing the dimensionality**   A simple method for altering the dimension of a flow is to take the output of an intermediate layer $z'$ and partition it into two pieces $z' = \{z'_1, z'_2\}$. Multi-scale architectures [6] match $z'_2$ directly to a base density and apply further transformations $z'_1$. Funnels [15] generalize this by allowing $z'_2$ to depend on $z'_1$, i.e. they work with the model $p(z') = p(f'(z'_1))p(z'_2|f'(z_1))|\det J_{z'_1}^{f'}|$ where the conditional distribution $p(z'_2|f'(z_1))$ is trainable. It is useful to think of these factorization schemes as dimension reducing mechanisms from $\dim(z')$ to $\dim(z'_1)$.

**Increasing the dimensionality**   Dimension increasing flow layers can improve a models flexibility, as demonstrated by augmented normalizing flows [2]. To increase the dimensionality, $x$ is embedded into a larger dimensional space and data independent noise $u$ is added to the embedding to obtain a distribution with support of nonzero measure. This noise addition $x \mapsto (x, u)$ is similar to dequantization [6, 22], but is orthogonal to the distribution of $x$ and increases its dimension from $\dim x$ to $\dim x + \dim u$. Under an augmentation the likelihood of $x$ can be estimated using

$$\log p(x) = \log \int du\, p(x, u) \tag{1}$$

$$= \log \int du\, \frac{p(u)p(x, u)}{p(u)} \tag{2}$$

$$\geq \int du\, p(u) \log \frac{p(x, u)}{p(u)} \tag{3}$$

$$= \mathbb{E}_{u \sim p(u)}\big[\log \frac{p(x, u)}{p(u)}\big] \tag{4}$$

$$= \mathbb{E}_{u \sim p(u)}\big[\log p(x, u) - \log p(u)\big]. \tag{5}$$

In practice, we estimate this expectation value by sampling $u$ everytime a datapoint is passed through the network. This means the integral is estimated with a single sample as in surVAEs [1].

# 3 Flowification

Suppose $\mathcal{A}$ is a network architecture with parameter space $\Theta$. Then for any choice of $\theta \in \Theta$ the network with parameters $\theta$ realizes a function $\mathcal{A}_\theta : \mathbb{R}^D \to \mathbb{R}^C$ for some $D$ and $C$. Similarly, a normalizing flow model $\mathcal{F}$ is a parametric distribution on some $\mathbb{R}^E$, where for any choice of $\gamma$ from their parameter space $\Gamma$ they define a density function $\mathcal{F}_\gamma$ on $\mathbb{R}^E$. In this work we show that a large class of neural network architectures can be thought of as a flow model by constructing a map

$$\left\{ \begin{array}{c} \text{network} \\ \text{architectures} \end{array} \right\} \to \left\{ \begin{array}{c} \text{flow} \\ \text{models} \end{array} \right\}. \tag{6}$$

The embedding of $\mathcal{A}$ to its flowification $\mathcal{F}^{\mathcal{A}}$ results in a flow model that can realize density functions on the augmented space $\mathbb{R}^D \times \mathbb{R}^N$ for some $N \geq 0$, which in turn induces a density on $\mathbb{R}^D$ by integrating out the component on $\mathbb{R}^N$. The parameter space of $\mathcal{F}^{\mathcal{A}}$ factorises as $\Theta \times \Phi$ where $\Theta$ is the parameter space of $\mathcal{A}$ and also that of the forward pass of $\mathcal{F}^{\mathcal{A}}$, while $\Phi$ parametrises the inverse pass of $\mathcal{F}^{\mathcal{A}}$. In the simplest case $\Phi = \emptyset$, i.e. flowification does not require additional parameters. It is in this sense that we claim that a large fraction of machine learning models are normalizing flows.

**Terminology**  In what follows we work with conditional distributions such as $p(z|x)$ and it will be practical to think of them as "stochastic functions" $p : x \mapsto z$, that take an input $x$ and produce an output $z \sim p(z|x)$. Conversely, we think of a function $f : x \mapsto z$ as the Dirac $\delta$-distribution $f(z|x) = \delta(z - f(x))$. These definitions allow us to have a unified notation for deterministic and stochastic functions such that we can talk about them in the same language. Consequently, when we say "stochastic function", it will include deterministic functions as a corner case. Depending on whether $f$ and $f^{-1}$ are deterministic or stochastic, we talk about left, right or two-sided inverses. We will be careful to be precise about this.

**Method**  In the following we consider the standard building blocks of machine learning architectures and enrich them by defining (stochastic-)inverse functions and calculating the likelihood contribution of each layer. Treating each layer separately allows density estimation models to be built through composition [1]. The stochastic inverse can use the funnel approach, which increases the parameter count, or the multi-scale approach, which does not. For simplicity we will only consider conditional densities in the inverse as this is more general, though it is not required. We will refer to this process as *flowification* and the enriched layers as *flowified*; non-flowified layers will be called *standard layers*. Flowified layers can then be seen as simultaneously being

- **Flow** layers that are invertible, their likelihood contribution is known and therefore can be used to train the model to maximize the likelihood.

- **Standard** layers that can be trained with losses other than the likelihood, but for which the likelihood can be calculated after this training with fixed weights in the forward direction.

## 3.1 Linear Layers

Let $L_{W,b} : \mathbb{R}^n \to \mathbb{R}^m$ denote the the linear layer of a neural network with parameters defined by a weight matrix $W \in \mathbb{R}^{m \times n}$ and bias $b \in \mathbb{R}^m$. Formally, $L_{W,b}$ is defined as the affine function

$$x \mapsto L_{W,b}(x) := Wx + b \qquad x \in \mathbb{R}^n. \tag{7}$$

**Definition 1.** *Let $\phi(z|x) : \mathbb{R}^n \to \mathbb{R}^m$ be a stochastic function. We say that $\phi$ is **linear in expectation** if there exists $W \in \mathbb{R}^{m \times n}$ and $b \in \mathbb{R}^m$ such that for any $x \in \mathbb{R}^n$ the expected value of $\phi$ coincides with the application of $L_{W,b}$*

$$\mathbb{E}_{z \sim \phi(z|x)}[z] = L_{W,b}(x). \tag{8}$$

*Similarly, we say that a stochastic function $\psi(z|x)$ is convolutional in expectation if the deterministic function $x \mapsto \mathbb{E}_{z \sim \psi(z|x)}[z]$ is a convolutional layer.*

In this section we *flowify* linear layers, by which we mean we construct a pair of stochastic functions, a forward $\mathcal{L}(z|x) : \mathbb{R}^n \to \mathbb{R}^m$ and an inverse $\mathcal{L}^{-1}(x|z) : \mathbb{R}^m \to \mathbb{R}^n$ such that the forward is linear in expectation and is compatible with the inverse in a way that will be made precise in the following paragraphs.

**SVD parametrization**

To build a flowified linear layer, the first step is to parametrize the weight matrices by the singular value decomposition (SVD)[23]. This involves writing $W \in \mathbb{R}^{m \times n}$ as a product $W = V\Sigma U$, where $U \in \mathbb{R}^{n \times n}$ is orthogonal, $\Sigma \in \mathbb{R}^{m \times n}$ is diagonal and $V \in \mathbb{R}^{m \times m}$ is orthogonal. This parametrization is particularly useful for our purposes because the orthogonal transformations are easily invertible and do not contribute to the likelihood, and the non-invertible piece of the transformation is localized to $\Sigma$.

**Parametrizing $U$ and $V$** We generate elements of the special orthogonal group $SO(d)$ by applying the matrix-exponential to elements of the Lie algebra $\mathfrak{so}(d)$ of skew-symmetric matrices. We parametrize $\mathfrak{so}(d)$ and perform gradient descent there. As the Lie-algebra is a vector space, this is significantly easier than working directly with $SO(d)$. See Appendix G for details.

**Parametrizing $\Sigma$** The matrix $\Sigma$ is of shape $m \times n$ containing the singular values on the main diagonal. We ensure maximal rank of $\Sigma$, by parameterizing the logarithm of the main diagonal, this way all singular values are greater than 0.

It is important to note that this parametrization is not without loss of generality. In particular, it does not include matrices of non-maximal rank nor orientation reversing ones, where either $U \in O(n) \setminus SO(n)$ or $V \in O(m) \setminus SO(m)$. This implementation detail does not change the general perspective we provide of linear layers as normalizing flows, but instead simplifies the implementation of flowified layers.

**Reducing the dimensionality**

**Definition 2.** *We call the tuple $(\mathcal{L}(z|x), \mathcal{L}^{-1}(x|z))$ a **dimension decreasing flowified linear layer** if $\mathcal{L}$ is dimension decreasing, linear in expectation and the following conditions are satisfied*

*(i) The forward is deterministic, given by $\mathcal{L}(z|x) = L_{W,b}(x)$,*

*(ii) The layer is right-invertible, $\mathcal{L} \circ \mathcal{L}^{-1} = id_z$,*

*(iii) The likelihood contribution of $\mathcal{L}$ can be exactly computed.*

To flowify dimension decreasing linear layers, we define the forward function $\mathcal{L}$ as a standard linear layer with parameters $W$ and $b$,

$$\mathcal{L}(z|x) = \delta(z - L_{W,b}(x)). \tag{9}$$

Since $W$ is parametrized by the SVD decomposition, $W = V\Sigma U$, we need to invert $V, U$ and $\Sigma$ separately. As $V$ and $U$ are rotations, they are invertible in the usual sense. To construct a stochastic inverse to $\Sigma$, we think of it as a funnel [15] and use a neural network $p_{inv}((Ux)_{(m:)}|\Sigma U x)$ that models the $n - m$ dropped coordinates as a function of the $m$ non-dropped coordinates. Again, this is not required to calculate the likelihood under the model, even a fixed distribution could be used, but introducing some trainable parameters significantly improves the performance of the flow that is defined by the layer. We use $\Sigma^{-1}$ to denote this stochastic inverse to $\Sigma$. The stochastic inverse function $\mathcal{L}^{-1}$ can then be written as

$$\mathcal{L}^{-1}(x|z) = U^T \circ \Sigma^{-1} \circ V^T (z - b). \tag{10}$$

Since the rotations don't contribute to the log-likelihood, the likelihood of data under a dimension decreasing flowified linear layer is

$$\log p(x) = \log p_{inv}((Ux)_{(m:)}|\Sigma U x) + \log \Sigma + \log p(z), \tag{11}$$

where $\log \Sigma$ denotes the sum of the logarithms of the diagonal elements of $\Sigma$.

**Theorem 3.** *The above choices for $\mathcal{L}$ and $\mathcal{L}^{-1}$ define a dimension decreasing flowified linear layer.*

*Sketch of proof.* The definition of the forward pass (9) makes the forward pass linear in expectation and satisfies (i) by definition. Unpacking the definitions and decomposing $W$ into its SVD form yields right-invertibility (ii) which in turn implies that the likelihood contribution can be exactly computed (iii). $\square$

When the inverse density is not made to be conditional the above ideas can be visualized as a standard multi-scale flow architecture [6] as shown in Fig. 1.

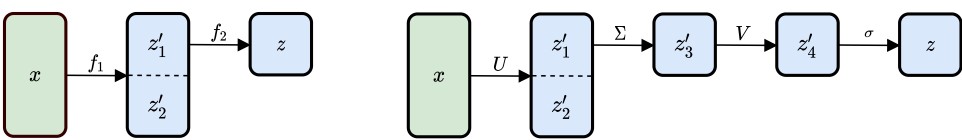

Figure 1: A multi-scale flow with a base density $p(z, z_2')$ on the left. A dimension reducing linear layer with activation $\sigma$ as a multi-scale flow with a base density $p(z, z_2')$ on the right.

**Increasing the dimensionality**

**Definition 4.** *We define the **Moore-Penrose pseudoinverse** $L_{W,b}^+$ of a linear layer $L_{W,b} : \mathbb{R}^n \to \mathbb{R}^m$ as the affine transformation $\mathbb{R}^m \to \mathbb{R}^n$*

$$z \mapsto L_{W,b}^+ := W^+(z - b) \qquad z \in \mathbb{R}^m \tag{12}$$

*where $W^+$ denotes the Moore-Penrose pseudoinverse of the matrix $W$.*

**Definition 5.** *We call the tuple $(\mathcal{L}(z|x), \mathcal{L}^{-1}(x|z))$ a **dimension increasing flowified linear layer** if $\mathcal{L}$ is dimension increasing, linear in expectation and the following conditions are satisfied*

    *(iv) The inverse $\mathcal{L}^{-1}$ is deterministic, given by $\mathcal{L}^{-1}(x|z) = L_{W,b}^+(z)$,*

    *(v) The layer is left-invertible, $\mathcal{L}^{-1} \circ \mathcal{L} = id_x$,*

    *(vi) The likelihood contribution of $\mathcal{L}$ can be bounded from below.*

To construct dimension increasing flowified linear layers, we rely again on the SVD parametrization where the only nontrivial component is $\Sigma$. In this case $\Sigma$ is a dimension increasing operation and we think of it as an augmentation step [2] composed with diagonal scaling. To augment, we sample $m - n$ coordinates from a distribution $p(u)$ with zero mean and then apply a scaling in $m$ dimensions. The likelihood contribution is then given by

$$\log p(x) \geq \mathbb{E}_{u \sim p(u)}\big[\log p(z) - \log p(u)\big] + \log \Sigma, \tag{13}$$

$$\log p(x) = \log \Sigma + \log p(z), \tag{14}$$

where $\log \Sigma$ denotes the sum of the logarithms of the $m$ scaling parameters. The inverse function $\mathcal{L}^{-1}$ is the composition of the inverse rotations, the inverse scaling and the dropping of the sampled coordinates. This sequence of steps is visualized in Fig. 2.

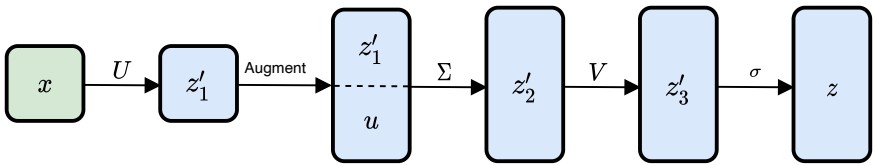

Figure 2: A dimension increasing flowified linear layer.

**Theorem 6.** *The above choices for $\mathcal{L}$ and $\mathcal{L}^{-1}$ define a dimension increasing flowified linear layer.*

*Sketch of proof.* The augmentation step [2] results in a lower bound on the likelihood contribution, implying (vi). Since $\mathbb{E}_{u \sim p(u)} = 0$, the augmentation does not influence the expected value of $z$, i.e. the forward pass is linear in expectation. Simple calculations using the SVD decomposition then imply both (iv) and (v). $\qquad\square$

**Preserving the dimensionality**

Dimension preserving layers are a corner case of both of the above scenarios, where padding and sampling are not needed in either direction and the layer is non-stochastically invertible. All this implies (i),(ii),(iii),(iv) and (v) are satisfied.

## 3.2 Convolutional layers

Convolutions can be seen as a dimension increasing coordinate repetition followed by a matrix multiplication with weight sharing. In the previous section we derived the specifics of matrix multiplication. We begin this section with the details of coordinate repetition after which we put the pieces together to build a flowified convolutional layer. In Appendix H we describe an alternative approach relying on the Fourier transform.

**Repeating coordinates** In this paragraph we focus on the $N$-fold repetition of a single scalar coordinate $x$. This significantly simplifies notation, but the technique generalizes in an obvious way. Intuitively, the idea is to expand the one dimensional volume to $N$ dimensions by first embedding and then increasing the volume of the embedding such that the volume in the $N-1$ directions complementary to the embedding can be controlled.

We have seen in §2 that the operation

$$x \mapsto (x, \underline{\mathbf{u}}) \qquad \underline{\mathbf{u}} = (u_1, ..., u_{N-1}), \tag{15}$$

has likelihood contribution $\mathbb{E}_{\underline{\mathbf{u}}}[-\log p(\underline{\mathbf{u}})]$. Now, we can apply any $N$-dimensional rotation $R_N$ which maps $(1, \underline{\mathbf{0}})$ to $\frac{1}{\sqrt{N}}(1, \underline{\mathbf{1}})$ to obtain[2]

$$R_N(x, \underline{\mathbf{u}}) = R_N(x, \underline{\mathbf{0}}) + R_N(0, \underline{\mathbf{u}}) = \frac{1}{\sqrt{N}}(x, \underline{\mathbf{x}}) + R_N(0, \underline{\mathbf{u}}). \tag{16}$$

Note that this rotation does not contribute to the likelihood. Finally, we apply a diagonal scaling in $N$ dimensions with factor $\sqrt{N}$ such that

$$x \mapsto (x, \underline{\mathbf{x}}) + R_N(0, \sqrt{N}\underline{\mathbf{u}}), \tag{17}$$

where the final scaling has likelihood contribution $N \log(\sqrt{N}) = (N/2) \log N$ and $x$ is now repeated $N$ times. The overall contribution to the likelihood of the embedding (17) is

$$\mathcal{V}(x, z) = \mathbb{E}_{\underline{\mathbf{u}}}[-\log p(\underline{\mathbf{u}})] + (N/2) \log N. \tag{18}$$

By construction, the padding distribution $R_N(0, \sqrt{N}\underline{\mathbf{u}})$ is orthogonal to the diagonal embedding $x \mapsto (x, ..., x)$ of the data distribution. The inverse function is given by the projection to the diagonal embedding,

$$(z_1, ..., z_N) \mapsto \frac{1}{N} \sum_i z_i. \tag{19}$$

**General architectures** Now that the likelihood contribution of arbitrary linear layers and coordinate repetition has been computed it is possible to flowify more general architectures such as convolutions and residual connections. It is important to note that just because an architecture works well for certain tasks, it is not clear if its flowified version will perform well at density estimation.

**Decomposing convolutional layers** To flowify convolutional layers, we decompose it as a sequence of building blocks that are easy to flowify separately. A standard convolutional layer performs the following sequence of steps:

1. **Padding** of the input image with zeros to increase its size.

2. **Unfolding** of the padded image into tiles. This step replicates the data according to the kernel size and stride.

3. **Applying a linear layer.** Finally, we apply the same linear layer to each of the tiles produced in the previous step. The outputs then correspond to the pixels of the output image.

---

[2] $\underline{\mathbf{0}}$ and $\underline{\mathbf{1}}$ denote the $(N-1)$-dimensional vectors $(0, ...0)$ and $(1, ..., 1)$, respectively. Similarly $\underline{\mathbf{x}}$ denotes the $(N-1)$-dimensional vector $(x, ..., x)$.

**Flowification**  Steps 1 and 3 are already flowified, i.e. their likelihood contribution is computed and an inverse is constructed, in §3.1. We denote their flowification with Pad and Linear, respectively. Step 2 fits into the discussion of the previous paragraph of repeating coordinates, where both its inverse (19) and its likelihood contribution (18) are given. We will denote this operation by Unfold.

**Definition 7.** *Let* Linear, Unfold *and* Pad *be as above and define* $\mathcal{C}$ *and* $\mathcal{C}^{-1}$ *be the following stochastic functions*

$$\mathcal{C} = \text{Linear} \circ \text{Unfold} \circ \text{Pad} \qquad (20)$$

$$\mathcal{C}^{-1} = \text{Pad}^{-1} \circ \text{Unfold}^{-1} \circ \text{Linear}^{-1} \qquad (21)$$

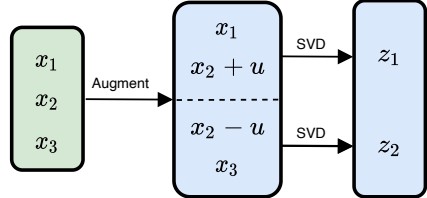

Figure 3: A flowified 1D-convolution with kernel size 2 applied to a vector with 3 features. The $x_2$ component appears in the operation twice, and so it is first duplicated so that the kernel can be applied to non-overlapping tiles.

*call the resulting layer* $(\mathcal{C}, \mathcal{C}^{-1})$ *a **flowified convolutional layer**.*

A flowified convolutional layer $(\mathcal{C}, \mathcal{C}^{-1})$ is then *convolutional in expectation* (Definition 1), i.e. there exists a convolutional layer $C_\theta$ with parameters $\theta$ such that

$$\mathbb{E}_{z \sim \mathcal{C}(z|x)}[z] = C_\theta(x). \qquad (22)$$

The flowification of a convolution without padding can be seen in Fig. 3. The Unfold operation is implemented as coordinate duplication and Linear is a flowified linear layer parameterized by the SVD.

**Activation functions**  Functions that are surjective onto $\mathbb{R}$ and invertible fit well in our framework as they can be used out of the box without any modifications. In our experiments we use LeakyReLU and rational-quadratic splines [7] as activations functions. Non-invertible activations can also be used when equipped with additional densities [1].

**Residual connections**  Residual connections can be seen as coordinate duplication followed by two separate computational graphs $\{f_1, f_2\}$ with the outputs recombined in a sum. The sum can be inverted by defining a density over one of the summands $p(f_1(x+u)|f_1(x+u), f_2(x-u))$ and sampling from this density, which will also define the likelihood contribution. Then, if the likelihood contribution can be calculated for each individual computational graph, the likelihood of the total operation can be calculated [1].

## 4  Experiments

To test the constructions described in the previous section we flowify multilayer perceptrons and convolutional architectures and train them to maximize the likelihood of different datasets.

**Tabular data**  In this section we study a selection of UCI datasets [24] and the BSDS300 collection of natural images [25] using the preprocessed dataset used by masked autoregressive flows [17, 26]. We compare the performance with several baselines for comparison in Table 1. We see that the flowified models have the right order of magnitude for the likelihood but are not competitive.

Table 1: Test log likelihood (in nats, higher is better) for UCI datasets and BSDS300, with error bars corresponding to two standard deviations.

| MODEL | POWER | GAS | HEPMASS | MINBOONE | BSDS300 |
|---|---|---|---|---|---|
| GLOW | $0.38 \pm 0.01$ | $12.02 \pm 0.02$ | $-17.22 \pm 0.02$ | $-10.65 \pm 0.45$ | $156.96 \pm 0.28$ |
| NSF | $0.63 \pm 0.01$ | $13.02 \pm 0.02$ | $-14.92 \pm 0.02$ | $-9.58 \pm 0.48$ | $157.61 \pm 0.28$ |
| FMLP | $-0.50 \pm 0.02$ | $5.35 \pm 0.02$ | $-19.56 \pm 0.04$ | $-14.05 \pm 0.48$ | $144.22 \pm 0.28$ |

**Image Data**  In this section we use the MNIST [27] and CIFAR10 [28] datasets with the standard training and test splits. The data is uniformly dequantized as required to train on image data [29, 30]. For both datasets we trained networks consisting only of flowified linear layers (FMLP) and also networks consisting of convolutional layers followed by dense layers (FCONV1). To minimize the number of augmentation steps that occur in each model we define additional architectures with similar numbers of parameters but with non-overlapping kernels in the convolutional layers (FCONV2). The exact architectures can be found in Appendix F.1. The flowified layers sample from $\mathcal{N}(0, a)$ for dimension increasing operations, where $a$ is a per-layer trainable parameter. We use rational quadratic splines [7] with 8 knots and a tail bound of 2 as activation functions, where the same function is applied per output node. We also ran experiments with coupling layers using rational quadratic splines [7] mixed into FCONV2, in all other cases the parameters of the model are not data-dependent operations. The improved performance of these models suggests that flowified layers do not mismodel the density of the data, but they do lack the capacity to model it well. Samples from these models are shown in Appendix C.

Table 2: Test-set bits per dimension (BPD) for MNIST and CIFAR-10 models, lower is better. Results from several other works were included for comparison. Flowified models with overlapping kernels FCONV1 and non-overlapping kernels FCONV2 are shown, with a similar parameter budget to the neural spline flow [7]. The models FCONV1 + NSF and FCONV2 + NSF correspond to architectures using rational quadratic spline layers in-between the flowified layers of FCONV1 and FCONV2, respectively. Samples from these models can be found in Appendix C.

| MODEL | MNIST | CIFAR-10 |
|---|---|---|
| GLOW [8] | 1.05 | 3.35 |
| REALNVP [6] | - | 3.49 |
| NSF [7] | - | 3.38 |
| I-RESNET [11] | 1.06 | 3.45 |
| I-CONVNET [14] | | 4.61 |
| MAF [17] | 1.91 | 4.31 |
| FMLP | 4.19 | 5.45 |
| FCONV1 | 3.11 | 4.91 |
| FCONV2 | 1.41 | 4.20 |
| FCONV1 + NSF | 2.70 | 3.93 |
| FCONV2 + NSF | 1.35 | 3.69 |

The results of the density modelling can be seen in Table. 2. The images in the left column of Fig. 4 are generated by sampling from a standard gaussian in the latent space and taking the expected output of the inverse in every layer. The images in the right column use the same latent samples as the left column but also sample from the distribution defined by the inverse pass of the layers.

As seen in Fig. 2 the FMLP models are outperformed by the FCONV models and the convolutional models with non-overlapping kernels achieve better results than the ones with overlapping kernels. This suggests both that the inductive bias of the convolution is useful for modelling distributions of images and that the augmentation step costs more in terms of likelihood than it provides in terms of increased expressivity.

## 5   Related Work

Several works have developed methods that allow standard layers to be made invertible, but these approaches restrict the space of the models, whereas we consider networks in their full generality. Invertible ResNets [11, 12] require that each residual block has a Lipschitz constant less than one, but even with this restriction they attain competitive results on both classification and density estimation. The same Lipschitz constraint can also be applied to other networks [13]. In these architectures the multi-scale architecture used in RealNVPs [6] is not leveraged, and so no information is discarded by the model. It is unclear why this approach outperforms flowified layers, as seen in Table. 2, but it could be due to the preservation of information through the model, the very large number of parameters that are used in these approaches, the restricted subspace of the models, or some combination of these three.

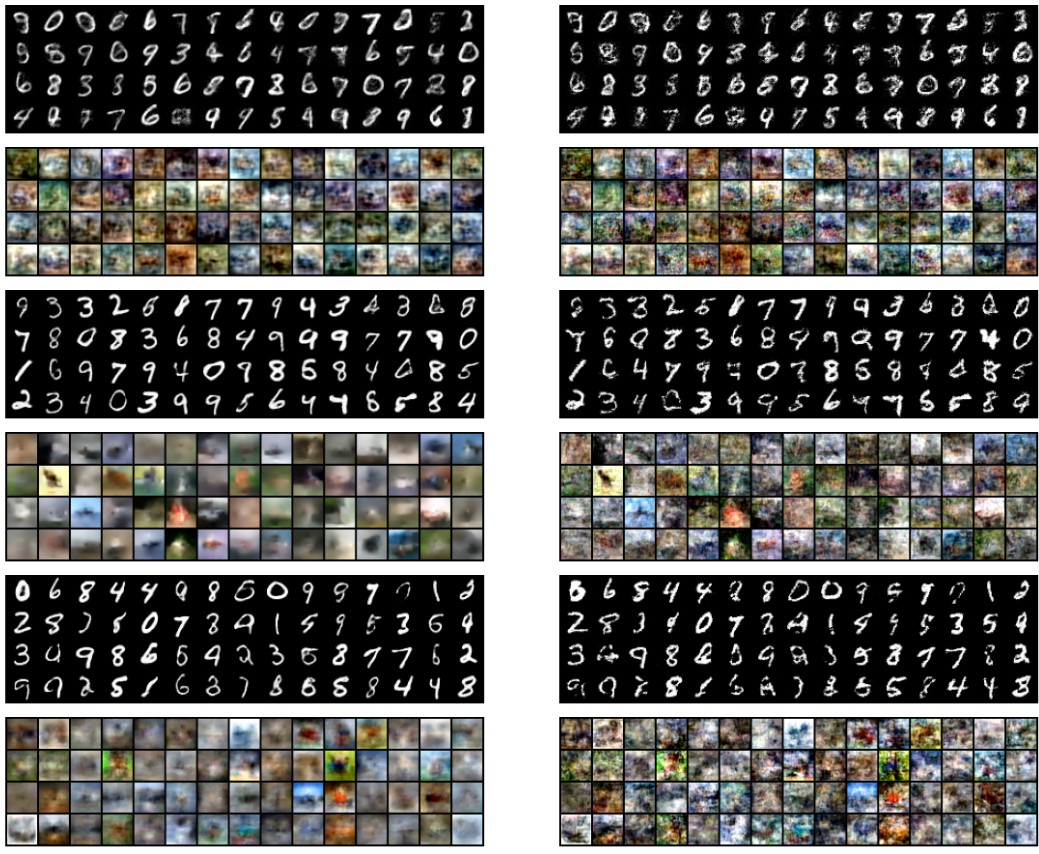

Figure 4: Samples from flowified multilayer perceptrons (top two rows) convolutional networks with overlapping (third and fourth rows) and non-overlapping (bottom two rows) kernels trained on MNIST and CIFAR-10. Samples where the mean is used to invert the SVD in the inverse pass (left column). Samples generated by drawing from the inverse density to invert the SVD in the inverse pass (right column).More samples from these models can be found in Appendix E.

It can also be shown that convolutions with the same number of input and output channels can be made invertible [14]. These layers perform poorly at the task of density estimation and are outperformed by flowified layers. This is likely due to the increased expressivity that comes from considering a larger space of architectures. There have been several works that develop convolution inspired invertible transformations [31–33], but these architectures consider restricted transformations to maintain invertibility.

## 6  Future Work and Conlusion

Our experiments suggest that flowified convolutional networks do not match the density estimation performance of similarly sized normalizing flows. A possible explanation is that the dimension reducing steps discard information and more expressive encoding layers are necessary to transform the distributions before reducing the dimensionality. This is supported by the experiments using NSF layers in-between the flowified layers (see App. C). The addition of NSF layers leads to improved performance both in terms of visual quality and also BPD values. Possibly the main limitation of a network consisting purely of flowified layers is the fact that, unlike flow layers, the forward passes of standard linear and convolutional layers are not data-dependent. This is reinforced by the fact that the entanglement capability typically used in flows also appears in attention mechanisms, which have been shown to excel at capturing complex statistical structures [34–36].

With further development – such as increased capacity given to the inverse density, or data dependent parameters in the forward pass – standard architectures could become competitive density estimators in their own right and allow for general purpose models to be developed. The focus of this work was on employing standard layers for density estimation, but it is possible that designing data dependent variants of standard layers that are more flow-like could improve their performance on tasks such as classification and regression. The flowification procedure provides a useful means for designing such models, and demonstrates that standard architectures can be considered a subset of normalizing flows, a correspondence that has not previously been demonstrated.

The code for reproducing our experiments is available under MIT license at `https://github.com/balintmate/flowification`.

## 7 Acknowledgement

The authors would like to acknowledge funding through the SNSF Sinergia grant called Robust Deep Density Models for High-Energy Particle Physics and Solar Flare Analysis (RODEM) with funding number CRSII5_193716.

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
