# OpenReview forum: "Flowification: Everything is a normalizing flow"
_NeurIPS.cc/2022/Conference — NeurIPS 2022 Accept_

### Official Review · Reviewer_rJok · 2022-07-05

**Rating:** 4
**Confidence:** 3
**Soundness:** 1 poor
**Presentation:** 2 fair
**Contribution:** 2 fair

**Summary:**

The paper introduces a method that transforms any multilayer perception into the flow model. The authors investigate various scenarios where they apply linear and convolutional layers as transformations inside the flow, either reducing or extending the dimensionality of the data. Some experiments on MNIST and CIFAR are performed to investigate the quality of the approach.

**Questions:**

I would like to ask the authors to refer to the remarks from the Weakness section, mainly focusing on:

1. Relation to SurVAE approach, how the surjective for inference or generation are designed, and how sampling for inference is performed.
2. The concerns about notations and the design choices (eq. (7), $\phi(z|x)$)
3. Relation to probabilistic PCA that uses the linear transformation.


**Ethics Review Area:**

["I don’t know"]

**Limitations:**

The limitations were included in the conclusion section.

**Strengths And Weaknesses:**

STRENGTHS

The defined problem and direction are important for the committee. The idea of searching for novel forms of transformations and connections with existing models seems to be attractive.

WEAKNESSES

I read the paper several times, and I had a problem identifying the contribution of this work. The title and the abstract suggest rather that we should expect to transfer standard MLPs into Bayesian networks using flows. In my opinion, the proposed solution rather enriches the family of transformations for SurVAE flows:

Nielsen, D., Jaini, P., Hoogeboom, E., Winther, O., & Welling, M. (2020). Survae flows: Surjections to bridge the gap between vaes and flows. Advances in Neural Information Processing Systems, 33, 12685-12696.

with linear and convolutional transformations. Assuming this, it is not clear if the proposed transformations are surjective for inference or generation. Assuming we consider the surjective inference, how the sampling process is implemented.

The paper is difficult to follow, the math transformations are difficult to follow, the notation seems to be unexplained. For instance:
- eq. (7) it is not clear what $p_{inv}$ is and why it is not expressed in log domain. What is $p(z)$ here.
- The choice of $\phi(z|x)$ is also not discussed in the paper.
- matrix $\Sigma$ should have positive values on diagonal I Guess.

The application of the linear layer is equivalent to the application of probabilistic PCA, where the likelihood can be easily calculated in the closed-form: $p(x)=\int p(x|z)p(z)dz$ can by calculated analytically.

There are no clear advantages of provided transformations compared to the proposed in SurVAE. Authors should discuss using the benefits of their approach compared to an existing one.

The clarity of writing and consistency with previous works regarding notation should be improved. Instead of \emph{Sketch of proof} the exact proof should be provided in the appendix, instead of above $\mathcal{L}$ the reference to the definition should be delivered. Moreover, using $\log \Sigma$ is further explained but practically means another type of operation.

My primary concern is about the experimental part. For such works, achieving SOTA is not crucial, but the results are far from baselines. There are also no toy examples that show the benefits of the approach for particular assumptions.

Concluding, the paper seems to be a bit of a work in progress and should be reorganized, and additional experimental evaluation should be included.

---

> ### Author Response · Authors · 2022-08-02
> **Response to Reviewer rJok**
>
> Response to rJok:
>
> We thank the reviewer for their comments and for recognizing the importance of the goal we set. Please note that there was an overlap between the reviews and we address these points together in a general comment to the paper, i.e. part of our response can be found there.
>
>
>
> *“The title and the abstract suggest rather that we should expect to transfer standard MLPs into Bayesian networks using flows. In my opinion, the proposed solution rather enriches the family of transformations for SurVAE flows”*
>
> In our opinion this work shows how standard MLPs can be described as normalizing flows, and that the method for doing so can be described using the SurVAE framework. It is unclear to us how these two things are incompatible.
>
> *“There are no clear advantages of provided transformations compared to the proposed in SurVAE. Authors should discuss using the benefits of their approach compared to an existing one”*
>
> As stated above, from our perspective, we demonstrate that the SurVAE framework can be applied to standard layers, resulting in a connection between architectures devised for fundamentally different tasks. Our methods are compared to several benchmarks in Tables 1, 2.
>
>
> *“Assuming this, it is not clear if the proposed transformations are surjective for inference or generation. Assuming we consider the surjective inference, how the sampling process is implemented”*
>
> Flowified dimension decreasing  operations are inference surjections in the terminology of the SurVAE paper. Similarly, flowified dimension increasing operations are generative surjections. Appendix A now describes the connection to the SurVAE framework in the updated version of the paper.
> We are unsure what the reviewer would like to know about how the sampling is implemented? A conditional density is trained such that it can be sampled from when generating new data points.
>
> *“eq. (7) it is not clear what $p_{inv}$ is and why it is not expressed in log domain. What is $p(z)$ here.“*
>
> $p_{inv}$ is the likelihood of the dropped coordinates conditioned on the non-dropped coordinates. You are absolutely correct, there was a typo, in the updated version of the paper it is expressed in the log domain. $p(z)$ is the likelihood in the “output space” of the layer.
>
> *“The choice of  $\phi(z|x)$ is also not discussed in the paper. “*
>
> The notation $\phi(z|x)$ is only used as a placeholder for a general stochastic function for the sake of stating the definitions. When defining the forward passes of linear and convolutional layers  $\phi(z|x)=\mathcal L(z|x)$ and  $\phi(z|x)=\mathcal C(z|x)$, respectively.
>
> *“The matrix $\Sigma$ should have positive values on diagonal I Guess.”*
>
> Indeed, the singular values are parametrized in the log domain, making the diagonal of  $\Sigma$ positive.
>
> *“The application of the linear layer is equivalent to the application of probabilistic PCA”*
>
> Probabilistic PCA seems to be similar to a flowified, dimension-decreasing linear layer without activation and a Gaussian prior, but there are significant differences that make the flowified linear layer much more expressive. Namely, a flowified linear layer can be equipped with a learnable inverse distribution that models the distribution in the kernel of the linear operator realized by the layer. This inverse need not be gaussian. Further, flowified linear layers can be composed together to learn a much more expressive map than probabilistic PCA.

---

> > ### Comment · Reviewer_rJok · 2022-08-07
> > **Post rebuttal comments.**
> >
> > Thank you for the rebuttal. Some of the concerns were clarified by the authors. I appreciate your effort in preparing the revised version of the manuscript, correcting the notation, and delivering additional results. It looks much better now. However, I am not sure how significant changes during rebuttal are allowed at this stage.
> >
> > I agree that empirical SOTA results are not crucial for such works. The question is, what is the reason to put such results into such work? I think it is better to focus on some particular benefits of the approach or present some educative use-cases that confirm theoretical aspects.
> >
> > The connection to Survae was also raised by reviewer LKEV. The authors still do not refer to the statement that their approach "enriches the family of transformations for SurVAE flows". Can we treat the MLP as a generalization of transformations of SurVAE? If yes, I expect a rather comprehensive evaluation of the transformations proposed in the Survae paper. Moreover, I will strongly suggest answering more to the point during the rebuttal. Jumping between the revised version and other responses is challenging for the reviewer, taking into account that the reviewer does not have the requirement to look into the manuscript and monitor each of the changes.
> >
> > Despite those doubts, I will raise my score, but only to 4.

---

> > > ### Author Response · Authors · 2022-08-08
> > > **Post rebuttal comments.**
> > >
> > > Thank you for the comments and taking the time to go through the rebuttal, in particular on how to make the rebuttal better.
> > >
> > > The purpose of the experiments was to validate that the theory made sense and the method could perform density estimation and sampling on data. The benchmarks we studied are standard for these tasks and we think this is the most canonical way to do the evaluation.
> > >
> > > In response to your question:
> > >
> > > ```angular2html
> > > Can we treat the MLP as a generalization of transformations of SurVAE?
> > > ```
> > > No, we do not think so. From our perspective the SurVAE is a framework (and this is how it is presented in the SurVAE paper [15]), not a set of transformations. All of the transformations presented with the SurVAE framework [15] can be composed with flowified layers in the same way the rebuttal composes flowified layers with rational quadratic flow layers. With this perspective we do not think that it makes sense to do an in depth comparison.
> > >
> > > [15] Didrik Nielsen, Priyank Jaini, Emiel Hoogeboom, Ole Winther, and Max Welling. Survae flows: Surjections to bridge the gap between vaes and flows. *Advances in Neural Information Processing Systems*, 33:12685-12696, 2020.

---

### Official Review · Reviewer_LKEV · 2022-07-12

**Rating:** 5
**Confidence:** 4
**Soundness:** 3 good
**Presentation:** 3 good
**Contribution:** 3 good

**Summary:**

The paper claims to provide a method to convert any convolutional or fully-connected layer into a normalizing flow. This is primarily performed by converting the representation into an equivalent SVD-representation using a pair of learnable orthonormal matrices(/tensors) in conjunction with a positive, diagonal matrix. This allows any NN composed of conv/linear layers to be converted to a NF and trained via MLE, possibly in conjunction with some other objective.

**Questions:**

Please see above.

**Limitations:**

Please see above.

**Strengths And Weaknesses:**

Originality: Are the tasks or methods new? Is the work a novel combination of well-known techniques? (This can be valuable!) Is it clear how this work differs from previous contributions? Is related work adequately cited?
The paper is heavily reliant on SURVAEs and does cite the relevant work. However, given the relative importance of the method it would be helpful if the paper explained this idea, possibly in an Appendix instead of just referencing the work. I found much of the paper confusing until I read this preceding paper.
The paper relies heavily on the use of the SVD reparameterization method but does not cite any of the preceding efforts that explore this idea. Some comment on the use of the skew-symmetric -> matrix exponentiation formulation of the orthogonal weights vs the more typical Househoulder reflection method should probably be discussed as well.

Quality: Is the submission technically sound? Are claims well supported (e.g., by theoretical analysis or experimental results)? Are the methods used appropriate? Is this a complete piece of work or work in progress? Are the authors careful and honest about evaluating both the strengths and weaknesses of their work?
The paper claims that their method converts any MLP/CNN into a NF. However, in fact, their method only produces a tractable likelihood estimate when the layer is contractive in dimension (or when the layer is dimension preserving). When the layer is dimension expanding, the resultant "NF" is only capable of providing an ELBO instead of an exact likelihood estimate. "Everything" isn't a NF unless you can (exactly) marginilize out the introduced random variables at the inlet of the dimension expanding layers. Since a lower bound on log-likelihood is an upper bound on negative log-likelihood and, therefore, BPD, it is unsurprising that the experimental results show much higher "BPD" than alternative NF which do provide a likelihood estimate and not a bound.

Clarity: Is the submission clearly written? Is it well organized? (If not, please make constructive suggestions for improving its clarity.) Does it adequately inform the reader? (Note that a superbly written paper provides enough information for an expert reader to reproduce its results.)
Most of the paper is accessible and well written. The discussion of padding and unfolding was somewhat confusing but could be understood on careful read.

Significance: Are the results important? Are others (researchers or practitioners) likely to use the ideas or build on them? Does the submission address a difficult task in a better way than previous work? Does it advance the state of the art in a demonstrable way? Does it provide unique data, unique conclusions about existing data, or a unique theoretical or experimental approach?
The general idea could be very important but I am skeptical that the proposed solution is as effective as presented. The method also claims to work for MLPs and CNNs but the experiments only demonstrate CNNs. The application of the idea to a toy dataset or the standard UCI likelihood datasets (power, gas, miniboone, hepmass, bsds) could be very informative.

General comments:
I really like the goal of the paper but am unconvinced that the paper achieves its claims, largely because of the limiation of dimension expanding operations. Perhaps I have misunderstood this operation and the conversion from a tractable likelihood estimator to a bound. If you can soften your claims or demonstrate my misunderstanding I am happy to increase my score. Similarly, if you can demonstrate your method on non-image datasets (possibly in an Appendix) I will reconsider my score.

---

> ### Author Response · Authors · 2022-08-02
> **Response to Reviewer LKEV**
>
> We thank the reviewer for the detailed feedback and for finding the general idea important. Please note that there was an overlap between the reviews and we address these points together in a general comment to the paper, i.e. part of our response can be found there.
>
> *“ The paper is heavily reliant on SURVAEs and does cite the relevant work. However, given the relative importance of the method it would be helpful if the paper explained this idea, possibly in an Appendix.”*
>
> To address this we have added a section on SurVAE in Appendix A. Rather than describe the method in depth we detailed how the flowification procedure fits into the surVAE framework and hopefully made the connection between surVAE and flowification clearer.
>
> *"The paper relies heavily on the use of the SVD reparameterization method but does not cite any of the preceding efforts that explore this idea. Some comment on the use of the skew-symmetric -> matrix exponentiation formulation of the orthogonal weights vs the more typical Househoulder reflection method should probably be discussed as well."*
>
> The matrix exponential has the advantage of generating the whole rotation group, while many Householder transformations need to be composed to have the same level of expressivity. This, of course, comes with an increased computational cost (scales quadratically) . We did not explore Householder transformations, but we expect that for linear layers with large input or output dimensionality, the matrix exponential will simply become too expensive and it will become necessary to rely on Householder transformations. We also added this paragraph to appendix G. We also added a citation for SVD reparametrization.
>
> *“When the layer is dimension expanding, the resultant "NF" is only capable of providing an ELBO instead of an exact likelihood estimate.”*
>
> We agree that there is a subtle terminology issue. From our perspective, even though the augmentation is dealt with by adding noise, the modeling of the resulting joint is done with a normalizing flow. In this respect it seems reasonable to call it a Normalizing Flow, although indeed the likelihood of the non-augmented data can be only lower-bounded. A discussion of this has been included in Appendix A. We would be interested to know if you think this is still unjustified.
>
> We feel further justified in our choice of terminology given that it has already been used in the augmented normalizing flows paper
>
> [13]  Chin-Wei Huang, Laurent Dinh, and Aaron Courville. Augmented normalizing flows: Bridging the gap between generative flows and latent variable models, 2020.
>
> The way we perform convolutions (pad, unfold, apply kernel) requires a large number of dim-expanding steps (unfold) and it is inevitable to work with this (weakened) version of NFs. If a convolutional layer is such that the dimensionality (channels times height times width) of its input is less than or equal than that of its output, the overall convolution operation is still in the dimension reducing regime, theoretically allowing to compute the exact likelihood. We investigated this approach by diagonalising the convolution using the Fourier transform, which unfortunately turned out to be too expensive computationally. We discuss this approach in Appendix H.
>
> Since a lower bound on log-likelihood is an upper bound on negative log-likelihood and, therefore, BPD, it is unsurprising that the experimental results show much higher "BPD" than alternative NF which do provide a likelihood estimate and not a bound.
>
> The augmented normalizing flow paper [13] demonstrates very good results on a range of datasets, and so we do not believe the modeling gap alone is the cause of the poor performance. Further, we have improved the performance of our models by considering a different type of noise in the augmentation.
>
> *“The method also claims to work for MLPs and CNNs but the experiments only demonstrate CNNs. The application of the idea to a toy dataset or the standard UCI likelihood datasets (power, gas, miniboone, hepmass, bsds) could be very informative.”*
>
> The architecture to which we only referred to as convolutional networks contain both convolutional and linear layers (the architectures are shown in the appendix). Since submission we also trained networks containing only flowified linear layers on MNIST, CIFAR10 and the UCI datasets. These results are now included in the updated version of the paper.

---

> > ### Comment · Reviewer_LKEV · 2022-08-08
> > **Thanks for the thorough response and modifications**
> >
> > Bottom line: I will increase my score to a 5.
> >
> > Thank you for including the additional discussion about SurVAEs and references/discussion of SVD reparameterizations. Similarly, I appreciate the additional architectures and UCI datasets, I recognize that this is a lot of work in a short time and am impressed you turned it around so quickly. It is a bit disappointing that the performance on the UCI datasets is so low. This doesn't seem inconsistent with your original results on the image datasets, though it appears you have made encouraging improvements there. Will similar modifications improve the performance on the tabular datasets or is the lower performance still something of a mystery?
> >
> > Similarly, thank you for clarifying that the performance gap may not be due to the difference in llh vs elbo based on the Augmented Flow results. I appreciate the honesty.
> >
> > ---
> > Per the discussion about normalizing flows and elbos vs likelihoods. I am of the opinion that a "NF" produces a likelihood estimate over the input data and an "Augment NF" (ANF) produces a variational version based on noise injection. I would prefer a title and discussion that matches this (or a similar) definition; otherwise it is difficult to distinguish what a model produces (lh vs elbo) without explicitly investigating the architecture. However, I recognize that this is a discussion of syntax and, while helpful for clarity and discussion, is orthogonal to the paper's primary technical advances.

---

> > > ### Author Response · Authors · 2022-08-09
> > > **Post rebuttal comments**
> > >
> > > Thank you for your comments!
> > >
> > > *"Will similar modifications improve the performance on the tabular datasets or is the lower performance still something of a mystery?"*
> > >
> > > The  modification that helped in the case of image datasets has already been applied to the architectures producing these results. We do not have a clear understanding of the lower performance, but believe that incremental improvements are still possible with systematic hyper-parameter tuning.  We still think that the main bottleneck is that the forward pass of a flowified network is the same as that of a standard network (the same network, pre-flowification).  Flowification in its current form is a conceptually interesting connection, but not a recipe for building competitive normalizing flows.
> > >
> > > *"I would prefer a title and discussion that matches this (or a similar) definition; otherwise it is difficult to distinguish what a model produces (lh vs elbo) without explicitly investigating the architecture. However, I recognize that this is a discussion of syntax and, while helpful for clarity and discussion, is orthogonal to the paper's primary technical advances."*
> > >
> > > We agree that it is a very important point and to make our claims precise, we included this discussion in the updated version. Regarding the title, “Everything is either a normalizing flow or an augmented normalizing flow” or  "Everything is an (augmented) normalizing flow" would be more precise. This rebuttal actually initiated a passionate discussion among the authors about changing the title that unfortunately has not converged at this time.

---

### Official Review · Reviewer_PUhx · 2022-07-12

**Rating:** 6
**Confidence:** 3
**Soundness:** 3 good
**Presentation:** 3 good
**Contribution:** 3 good

**Summary:**

The paper proposes a general approach for making commonly used neural network layers (e.g. linear and convolutional) invertible and usable as construction blocks of normalizing flows. Notably, under some assumptions, these “flowified” layers are allowed to shrink or expand dimensionality of the inputs. The proposed building blocks are tested as parts of convolutional neural networks on the density modeling and generation tasks on the MNIST and CIFAR10 benchmarks.

**Questions:**

(See Strengths & Weaknesses)
* The authors propose several hypotheses for why their method performs worse compared to the conceptually similar i-ResNet (Section 5). Can additional experiments be provided in order to support one or more of these hypotheses?

**Limitations:**

Limitations of the proposed method are unclear due to the limited set of presented experiments and architectural differences between the authors’ model and the competing methods their model is compared against.

**Strengths And Weaknesses:**

*Strengths*
* The material is well-presented and the paper is clearly written. The main contribution (“flowification” procedure) is described in an accessible way despite its mathematical complexity.
* The paper contains novel theoretical contributions, namely the “flowification” procedure, that would be interesting to a broad audience.

*Weaknesses*
* Empirical evaluation of the proposed method is rather limited (just two datasets). The results are also somewhat underwhelming (far behind the presented baseline methods). But more importantly, as indicated by the authors, the cause of underwhelming empirical performance is unclear. Providing further insight into this through additional experiments would be very valuable.

---

> ### Author Response · Authors · 2022-08-02
> **Response to Reviewer PUhx**
>
> We thank the reviewer for the positive feedback about the clarity of the presentation and their opinion that the novel theoretical contribution we make would be interesting to a broad audience. Please note that there was an overlap between the reviews and we address these points together in a general comment to the paper, i.e. part of our response can be found there.
>
> *“Empirical evaluation of the proposed method is rather limited (just two datasets). "*
>
> See general comment.
>
> *"The results are also somewhat underwhelming (far behind the presented baseline methods). But more importantly, as indicated by the authors, the cause of underwhelming empirical performance is unclear. Providing further insight into this through additional experiments would be very valuable.”*
>
> See general comment.
>
> The boost in performance improvement observed by changing the noise that is used for the augmentation points to fine tuning of the architectures as being an important factor in the observed performance gap. The performance improvement that comes from including rational quadratic neural spline flow layers with flowified layers points to the need for the self modulation inherent in these layers.
> We think that the full clarification of the modeling gap would be a significant contribution in its own right.
>
> *“The authors propose several hypotheses for why their method performs worse compared to the conceptually similar i-ResNet (Section 5). Can additional experiments be provided in order to support one or more of these hypotheses?”*
>
> There are certainly experiments that could support or reject some of these hypothesis, but unfortunately we did not find the time to run them in time for the rebuttal deadline.

---

### Official Review · Reviewer_Kcoo · 2022-07-13

**Rating:** 7
**Confidence:** 2
**Soundness:** 4 excellent
**Presentation:** 4 excellent
**Contribution:** 3 good

**Summary:**

The authors present flowification techniques for linear and convolutional layers by proposing to see linear layer weight matrices as parametrized by SVD decomposition, and convolutions as coordinate repetition followed by matrix multiplication. This enables the likelihood contribution of these layers and the computation of their inverse-- i.e., flowify them.

**Questions:**

- Why are there no experiments on density modelling considered just for flowified linear layers? I am curious as to how they perform as compared to flow-based models.
- How does the flow for the flowified CNNs look visually? It would be interesting to compare the flows from FCONV with other state of the art models like i-ResNet etc.
- Also, wouldn't it make sense to also consider classification error experiments on MNIST and CIFAR-10 with non flow-based models included as baselines as well?

It is possible that I did not understand the experiment design properly, and am willing to adjust my score accordingly.

**Limitations:**

Yes, I think the authors adequately address the limitations and impact of the proposed flowification in terms of its performance and the possible causes for it, and how it enables us to see standard architectures as subsets of normalizing flows. However, it would also be great to see potential negative societal impact be addressed in the paper (if there is anything specific), which I don't see as being addressed in the current version.

**Strengths And Weaknesses:**

*Strengths*

- The proposed formulation for unifying general architectures and normalizing flows looks mathematically elegant and sound. It would make it possible to realise the benefits of flows for general layers, as the authors experimentally demonstrate as well.
- The paper is clear and well-written-- the intuition provided for the definitions and theorems is helpful to understand them.

*Weaknesses*

- There are experiments for flowified convolutional architectures, but I couldn't find any experiments for flowified linear layers. Wouldn't it make sense to see them as well?
- It would make it clearer to have a representational diagram for a dimension-increasing flowified linear layer as well to get an overview.
- Although it's interesting to see flowified CNNS, their performance is not at the level of flow-based models, which is recognised by the authors as well.

---

> ### Author Response · Authors · 2022-08-02
> **Response to Reviewer Kcoo**
>
> We thank the reviewer for ​​finding the formulation mathematically elegant and sound. Please note that there was an overlap between the reviews and we address these points together in a general comment to the paper, i.e. part of our response can be found there.
>
>  *“It would make it clearer to have a representational diagram for a dimension-increasing flowified linear layer as well to get an overview.”*
>
> We have added this diagram as Figure 4 to Appendix A.
>
> *“Why are there no experiments on density modelling considered just for flowified linear layers? I am curious as to how they perform as compared to flow-based models.”*
>
> We have trained networks containing only flowified linear layers on MNIST, CIFAR10 and the UCI datasets. These results are now included in the updated version of the paper in Tables. 1, 2.
>
> *“How does the flow for the flowified CNNs look visually? It would be interesting to compare the flows from FCONV with other state of the art models like i-ResNet etc. “*
>
> Could the reviewer please clarify what these statements mean? The flowified CNN architecture is contained in the appendix, and we were wondering in what way the comparison would be made to models like i-ResNet other than looking at the BPD values?
>
>
> *“However, it would also be great to see potential negative societal impact be addressed in the paper (if there is anything specific), which I don't see as being addressed in the current version.”*
>
> We do not think there is any specific negative societal impact that can be predicted from what we have developed. This probably falls under the same umbrella as convolutional networks in general.

---

> > ### Comment · Reviewer_Kcoo · 2022-08-08
> > **response to author rebuttal**
> >
> > Thanks to the authors for the clarifications and for updating the draft.
> >
> > As a general comment, it would be much easier to see edits in a different color than the rest of the text. That aside,
> >
> > - What I mean by seeing how the flows look visually  is through generative modeling experiments as seen in the invertible residual networks paper (Figure 2 here https://arxiv.org/pdf/1811.00995.pdf).
> > - Thanks for including Figure 4, and Tables 1 and 2.
> >
> > Given these, I think I would not lean towards increasing the current score of 7 for the paper.

---

### Author Response · Authors · 2022-08-02
**Response to all reviewers**

We thank the reviewers for their feedback. The two concerns that were raised by many of the reviewers are addressed in this comment. These are 1) the small number of experiments and 2) the fact that flowified convolutional networks perform below the baseline models. To address the other points we will respond in detail to each reviewer separately.


 ### *Additional experiments*:

Since the submission we trained networks containing only flowified linear layers on MNIST, CIFAR10 and the UCI datasets. These results are now included in the updated version of the paper.


### *Results far from baselines*:

Since the submission we have improved the performance of the networks by changing the augmenting noise from uniform to normal. Since the convolutional networks with overlapping kernels perform many augmentation steps, their performance has improved significantly due to this change. Table 2 was changed accordingly. The updated paper contains a discussion on the choice of the augmenting noise in Appendix B.
When flowified layers are combined with NSF layers, the performance gap can be made even smaller (see Table 2 in the updated version of the paper) .

Most importantly, our main goal is not to compete with high-performing flow models, but to point to an interesting connection between seemingly very different architectures. Our architectures only consist of flowified layers that have not benefited from years of hyper-parameter tuning on the task of density estimation. We actually find it satisfying that a convolutional network can be flowified and trained as a normalizing flow to achieve reasonable performance.

---

### Meta-Review · Area_Chair_PkcM · 2022-08-26

**Recommendation:** Accept
**Confidence:** Certain

**Metareview:**

This paper proposes a straightforward framework to adapt a wide class of DNNs to be amenable to build normalizing flow. The framework which is derived is neat and is mostly supported experimentally. While some reviewers pointed that the links with SURVAEs could be stated more clearly in the initial submission and noted some weaknesses in the experimental part, the authors have done a convincing rebuttal. It seems that the contribution of this work is significant enough to lead to acceptance.

**Award:**

No

---

### Decision · Program_Chairs · 2022-09-14

Accept